# The Effects of Multidisciplinary Team Meetings on Clinical Practice for Colorectal, Lung, Prostate and Breast Cancer: A Systematic Review

**DOI:** 10.3390/cancers13164159

**Published:** 2021-08-18

**Authors:** Lejla Kočo, Harm H. A. Weekenstroo, Doenja M. J. Lambregts, J. P. Michiel Sedelaar, Mathias Prokop, Jurgen J. Fütterer, Ritse M. Mann

**Affiliations:** 1Department of Imaging, Radboud Institute for Health Sciences, Radboud University Medical Center, 6500 HB Nijmegen, The Netherlands; mathias.prokop@radboudumc.nl (M.P.); jurgen.futterer@radboudumc.nl (J.J.F.); ritse.mann@radboudumc.nl (R.M.M.); 2Department of Radiology, The Netherlands Cancer Institute, P.O. Box 90203, 1006 BE Amsterdam, The Netherlands; d.lambregts@nki.nl; 3Department of Urology, Radboud Institute for Health Science, Radboud University Medical Center, 6500 HB Nijmegen, The Netherlands; michiel.sedelaar@radboudumc.nl

**Keywords:** multidisciplinary team meeting, neoplasms, patient management, patient outcomes, survival

## Abstract

**Simple Summary:**

Multidisciplinary team meetings have increasingly been implemented in cancer care worldwide to ensure timely, accurate and evidence-based diagnosis, and treatment plans. Nowadays, multidisciplinary team meetings are generally considered indispensable. However, they are considered time-consuming and expensive, while the effects of multidisciplinary team meetings are not yet fully understood. The aim of this systematic review is to update and summarize the literature and create an overview of the existing knowledge. Cancer types such as colorectal, lung, prostate and breast cancer with rapidly increasing incidence rates will inevitably impact the workload of clinicians. Understanding the effects of the widely implemented multidisciplinary team meetings in oncology care is fundamental in order to optimize care pathways and allocate resources in the rapidly diversifying landscape of cancer therapies.

**Abstract:**

Objective: The aim of our systematic review is to identify the effects of multidisciplinary team meetings (MDTM) for lung, breast, colorectal and prostate cancer. Methods: Our systematic review, performed following PRISMA guidelines, included studies examining the impact of MDTMs on treatment decisions, patient and process outcomes. Electronic databases PUBMED, EMBASE, Cochrane Library and Web of Science were searched for articles published between 2000 and 2020. Risk of bias and level of evidence were assessed using the ROBINS-I tool and GRADE scale. Results: 41 of 13,246 articles were selected, evaluating colorectal (21), lung (10), prostate (6) and breast (4) cancer. Results showed that management plans were changed in 1.6–58% of cases after MDTMs. Studies reported a significant impact of MDTMs on surgery type, and a reduction of overall performed surgery after MDTM. Results also suggest that CT and MRI imaging significantly increased after MDTM implementation. Survival rate increased significantly with MDTM discussions according to twelve studies, yet three studies did not show significant differences. Conclusions: Despite heterogeneous data, MDTMs showed a significant impact on management plans, process outcomes and patient outcomes. To further explore the impact of MDTMs on the quality of healthcare, high-quality research is needed.

## 1. Introduction

Multidisciplinary care has increasingly been implemented in cancer care pathways worldwide, with oncology multidisciplinary team meetings (MDTMs) as a central platform for coordinated care delivery. An MDTM can be defined as “a group of healthcare professionals with different specialties who meet periodically (e.g., weekly) to discuss patient cases, diagnosis and treatment recommendations” [1]. MDTMs are often tailored as disease-specific and therefore differ in organization, i.e., in meeting frequency, duration or core team. The goal of the MDTM is to ensure timely, accurate and evidence-based diagnosis, treatment plans and follow-up for all discussed patients [2]. In 1995, Calman-Hine showed a positive correlation between multidisciplinary care and optimal decision making for cancer patients [3]. Since the publication of the Calman-Hine report, MDTMs have increasingly been adopted as part of routine cancer care pathways and are nowadays generally considered indispensable [2]. However, at the same time, MDTMs are considered time-consuming and expensive. The total workload of clinicians, occupied by MDTMs, is expected to increase, especially for cancer types with continuously increasing incidence rates, such as colorectal, lung, prostate and breast cancer [4].

These cancer types together constitute the top four cancer types in terms of global annual incidence [5]. Therefore, it is of great importance that the impact of MDTMs on different aspects of the clinical pathway and patient outcomes are well-understood. Previous systematic reviews showed weak evidence of impact on diagnosis and management plans. However, these studies found little evidence that MDTMs improve clinical outcomes [6,7,8]. We are the first to report on multiple cancer types in detail, to compare the effects of MDTMs in the four cancer types (colorectal, lung, prostate and breast cancer) that are expected to have a high impact on global healthcare. The aim of this systematic review is to update and summarize the literature and create an overview of the existing knowledge regarding the effects of MDTMs for colorectal, lung, prostate and breast cancer and to identify their value in these patient care pathways.

## 2. Methods

This systematic review was performed according to the Preferred Reporting Items for Systematic Reviews and Meta-Analyses (PRISMA) guidelines and Cochrane Collaboration’s double-data collection and extraction methodology [9,10].

### 2.1. Protocol and Registration

The protocol of this systematic review is registered in the PROSPERO database (CRD42019127476) [11].

### 2.2. Search Strategy

Relevant studies were searched in the following electronic databases: (1) PUBMED, (2) EMBASE, (3) Cochrane Library and (4) Web of Science. A librarian was consulted for the search strategy, and the search strategy combined variations for ‘multidisciplinary team meetings’ and ‘colorectal cancer’, ‘lung cancer’, ‘prostate cancer’ or ‘breast cancer’, (Appendix A). No language restrictions were applied. Time limits were from 1 January 2000 to 31 December 2020. In addition, reference lists of relevant systematic reviews were screened to identify additional studies.

### 2.3. Study Selection Criteria

The inclusion criteria consisted of randomized controlled trials (RCT), non-randomized controlled trials, observational studies and before-and-after studies. Typically, observational studies evaluated plans prior to and after MDTM discussion and before-and-after studies compared a cohort of patients that were discussed in MDTM with a control cohort.

Studies were included if the impact of MDTMs was examined for colorectal cancer, lung cancer, prostate cancer or breast cancer. Studies that evaluated mixed cohorts, such as urological cancers, were included if extracting specific data on prostate cancer was feasible. MDTMs were defined as “regular meetings where a multidisciplinary team of specialists attend and discuss diagnosis and treatment recommendations for patients” [1].

Studies that did investigate MDTMs for any of the four cancer types were critically evaluated and excluded if they met any of the following exclusion criteria. Study designs that were excluded were qualitative studies and studies without control groups. Studies were also excluded if they investigated: (1) the effect of another intervention, implemented in addition to the MDTM (e.g., telecommunication), (2) the implementation process, (3) opinions of healthcare professionals or (4) adherence to MDT advice.

### 2.4. Data Collection and Extraction

Abstracts of articles yielded from the search were imported in Endnote, and duplicates were removed [12]. Thereafter, the web application Rayyan QCRI was used for the screening process [13]. Two reviewers (L.K. and H.W.) independently performed a title–abstract screening, and articles that met the inclusion criteria were selected for full-text screening. Subsequently, the full-text articles were retrieved and reviewed (L.K. and H.W.). Disagreements on selection were discussed regularly and resolved by consensus. If no agreement could be reached, a third independent investigator (R.M.) could be consulted. Cohen’s kappa was calculated to determine the inter-observer variability of full-text screening. The data were extracted independently by two reviewers to ensure correct extraction (L.K. and H.W.). Meta-analysis in general was not possible due to the heterogeneity of the data, however a few parameters were calculated. A weighted average was calculated for changes in management plans per cancer type, weighing the percentage of changed plans with the number of included patients. Studies that did not perform statistics were not included in the analysis of data. Summative tables were created to present the data.

### 2.5. Quality Assessment

Two researchers (L.K. and H.W.) independently assessed the risk of bias, using the ROBINS-I tool for non-randomized studies [14]. In addition, the quality of evidence of the studies was assessed with the GRADE scale [15] by the same researchers (L.K. and H.W.). Any disagreements were first discussed between the reviewers, and if required, a third researcher (R.M.) could be consulted.

## 3. Results

A total of 20,912 articles were retrieved from electronic database searches (Figure 1). After removal of duplicates, 13,246 studies were evaluated in the title–abstract screening. Following title-abstract screening, 165 full-text articles were assessed for eligibility, and 41 studies were included in the final analysis. The measured Cohen’s Kappa for the inter-observer agreement between reviewers was 0.656, indicating substantial agreement [16]. It was not necessary to consult a third reviewer during title–abstract and full-text screening, as no disagreements remained after discussion.

### 3.1. Study Characteristics

Forty-one articles investigated the effect of MDTMs on cancer care pathways in colorectal cancer (21/41) [17,18,19,20,21,22,23,24,25,26,27,28,29,30,31,32,33,34,35,36,37], lung cancer (10/41) [38,39,40,41,42,43,44,45,46,47], prostate cancer (6/41) [48,49,50,51,52,53] and breast cancer (4/41) [54,55,56,57] (Table 1). All studies were conducted in adults (total *n* = 82,073, range 42–32,569 study subjects per study, mean 2002), with studies originating from in Oceania (8/41) [20,26,29,38,39,43,45,52], Asia (9/41) [18,21,23,35,41,46,50,56,57], Europe (13/41) [17,19,24,25,27,31,32,34,36,37,44,48,49], North America (10/41) [22,28,30,33,40,42,47,51,53,54] or Africa (1/41) [55]. Studies were performed in multicenter (6/41) [34,37,38,40,41,56] or single-center (35/41) [17,18,19,20,21,22,23,24,25,26,27,28,29,30,31,32,33,35,36,39,42,43,44,45,46,47,48,49,50,51,52,53,54,55,57] settings and according to hospital types: general hospitals (12/41) [19,24,25,29,32,36,38,42,46,48,54,55,57], university hospitals (16/41) [17,18,22,26,27,28,30,33,34,35,43,44,47,50,51,53], tertiary hospitals (8/41) [20,21,23,31,39,45,49,52] or unspecified (4/41) [37,40,41,56].

Overall, MDTM characteristics were similar across all included articles. In general, members of MDTMs include surgeons, (radiation) oncologists, radiologists, pathologists, residents and nurses. The frequency of MDTMs ranged from 4 times a week [21], to weekly [17,19,20,22,23,24,25,29,33,34,35,36,38,39,42,43,44,45,46,49,52,53,54,55] to biweekly [26,28,30,40,47,48]. Ten articles provided limited or no information on the frequency of MDTMs [18,27,31,32,37,41,50,51,56,57]. The outcomes measured in the articles were changes in or differences between: (1) process outcomes (9/41), (2) patient management (34/41) and (3) patient outcomes (18/41). Studies using a before–after study design (12/41) [17,23,24,26,28,32,34,35,36,44,46,55] or a case-control study design (13/41) [18,20,25,27,31,33,37,38,39,40,43,47,56] investigated a combination of all three outcome groups. When a cohort study design (16/41) [19,21,22,29,30,41,42,45,48,49,50,51,52,53,54,57] was used, most investigated patient management (13/16).

### 3.2. Risk of Bias and Quality of Evidence Assessment

With the use of the Robins-I tool, the risk of bias was rated critical in 22 studies (22/41) [17,18,19,20,21,22,23,25,26,27,30,31,32,33,34,35,42,44,46,49,50,53] and serious in 17 studies (17/41) [24,28,29,36,37,38,39,40,41,43,45,47,48,52,55,56,57]. Two studies showed no clear indication of serious or critical risk of bias but lacked information in one or more key domains of bias that hampered proper determination of overall risk of bias [51,54]. Based on the GRADE scale, the level of evidence had to be rated low in 2 studies [51,54], and very low in 39 studies [17,18,19,20,21,22,23,24,25,26,27,28,29,30,31,32,33,34,35,36,37,38,39,40,41,42,43,44,45,46,47,48,49,50,52,53,55,56,57] (Appendix A).

Nine studies assessed the effect of MDTMs on process outcomes. Reported process outcomes included time to treatment, time to diagnosis, costs and other process outcomes (Table 2).

For colorectal cancer, two studies showed significantly increased length of time to surgery in the MDTM cohort [26,37]. Chinai et al., estimated the annual costs of colorectal MDTMs, including direct and overhead costs, at £162,734 [19].

For lung cancer, Boxer et al., found a significantly increased time to start chemotherapy with palliative intent, while time to all other treatments remained unchanged [38]. Freeman et al., however showed a significantly shorter time to treatment when patients were discussed in the MDTM. They also assessed the mean cost of care and found that mean costs reduced significantly in the MDTM group compared to the control group. Furthermore, they also showed a significantly increased adherence to national guidelines and research participation [40]. Muthukrishnan et al. found that the MDTM group showed a significantly longer time to complete staging, time from imaging to diagnosis, staging to therapy and imaging to therapy. However, for a subgroup with stage I–III, only the time from staging to therapy was significantly longer in the MDTM group [47].

Brandão et al. showed that time from diagnosis to treatment for breast cancer patients was not significantly different between control and MDTM groups. Furthermore, they confirmed the cost-effectiveness of MDTM implementation [55].

Two articles reporting on process outcomes did not perform statistical analysis [29,43].

### 3.3. Patient Management

#### Changes in Overall Management Plans

In total, fourteen studies investigated the effect of MDTMs on overall management plans (treatment and/or diagnostic procedures) in a cohort study design. These studies compared definitive MDTM plans to those determined prior to MDTM case evaluation (Table 3) [19,20,21,22,30,42,45,48,49,50,51,52,53,54]. Overall, these studies reported an effect on management plans that were changed in 1.6–58% of all cases. In colorectal cancer, MDTMs affected overall management in 6–29% of the cases, resulting in a weighted average change of 16.2% [19,20,21,22,30], whereas lung cancer MDTMs changed management plans in 53.2% on average, ranging between 53% and 58% [42,45]. In prostate cancer, MDTMs changed management plans in 27.1% on average, ranging from 1.6–43% [48,49,50,51,52,53]. Breast cancer was only reported in one study, showing a total change of 42.1% of the management plans [54].

Nine studies investigated the type of changes. In six studies (all cancer types), researchers showed that treatment plans were changed more often (range 12.9–94.9%, mean 35.0%) than diagnostic plans (range 4–71.0%, mean 25.9%) [20,21,22,30,51,54]; in contrast, two studies (colorectal and lung cancer) showed the opposite effect [42,45]. One study on prostate cancer showed that changes in treatment plans occurred most often for patients with local disease, while changes in both diagnostic and treatment plans were more common in patients with advanced disease [49].

Six articles subdivided the whole cohort according to disease stage. For colorectal cancer (CRC) patients, one study showed that the percentage of management plans changed less often in newly diagnosed CRC cases (7.6%) compared to recurred CRC cases (16.4%) [21]. In prostate cancer, four articles reported changes in management plans subdivided according to cancer characteristics, presenting conflicting results. De Luca et al. showed that management plans changed more often in advanced (46.9%) and metastatic (33.4%) disease compared to local disease (23.2%) [49]. Similarly, Rao et al. showed changes in management plans in 23% of localized and 38% of metastatic prostate cancer cases [52]. El Khoury et al. and Kurpad et al., showed no trends when subdividing the patients according to Gleason score and disease stage, respectively [50,51]. Murthy et al., showed that changes were more common in early breast cancer patients (Stage 0–IIB: 8–27%) than in advanced patients (stage IIIA–IV: 0–7%) [54].

One article measured other outcomes in colorectal cancer patients, i.e., the type of MDT presentation (preoperative, follow-up, etc.) and whether the initial plan was tentative or definitive. Postoperative management plans (6.3%) changed significantly less often compared to initial (32.4%) or follow-up plans (35.2%). Tentative management plans (45.5%) changed significantly more than definitive plans (9.6%) [22].

### 3.4. Effect on Diagnostics and Treatment

Twenty-three studies investigated the effects of MDTMs on diagnostics and/or treatments using either a before–after or a case-control study design. In contrast to management plans, these studies focused on the diagnostics and/or treatments the patients actually received. These outcomes were not reported for prostate cancer.

#### 3.4.1. Diagnostics

Six studies investigated the effect of MDTMs on diagnostics received by colorectal cancer patients, i.e., MRI, CT, ultrasound or colonoscopies (Table 4). Of these six studies, three measured MRI imaging, and all showed a significant increase in the MDTM cohort, compared to the control group [17,20,36]. CT imaging was shown to increase significantly by four studies [17,20,35,37], and one study showed no significant effect [36]. In addition, Fernando et al., showed that the use of chest CT significantly increased, while CT of the abdomen did not change significantly in the MDTM group [20]. Three studies found no significant difference in ultrasound imaging [17,20,36]. Of the four studies investigating colonoscopies, three showed no significant difference between MDTM and control groups [20,28,36].

Two studies did not perform statistical analyses on (all) of these outcomes [17,27].

#### 3.4.2. Surgery

Fourteen studies investigated the preferred surgical type and whether surgery was performed (Table 4).

In colorectal cancer, all studies reporting on resection of primary tumors indicated a reduction in the MDTM cohort [23,37], and all studies reporting on surgical type showed a significant effect of MDTMs on the preferred surgical type [27,28,33]. Foucan et al., showed that colorectal cancer patients with advanced disease received different treatments depending on the MDTM status, unlike patients with early-stage disease. Stage III and IV patients without MDTM discussion most often received surgery alone, whereas patients in the MDTM group received mostly surgery followed by chemotherapy [37]. In lung cancer, Boxer et al., showed no significant effect on the number of performed surgeries [38], however Freeman et al., showed a significant reduction in (non-therapeutic) surgical procedures [40]. Tamburini et al., showed a significant effect on preferred surgical type [44]. For breast cancer, no significant increases in surgery or surgery type were identified [55].

#### 3.4.3. Radiotherapy, Chemotherapy and Palliative Care

Twelve studies investigated the effects of MDTMs on how many patients received chemotherapy, radiotherapy and palliative care (Table 4).

In colorectal cancer, Lan et al., showed a significant increase in the use of radiotherapy in the MDTM cohort [23], while MacDermid et al., showed no significant effect of MDTMs [24]. Similarly, in lung cancer, Boxer et al., showed a significant increase in the use of radiotherapy in the MDTM cohort [38], while Bydder et al., showed no significant effect of MDTMs [39]. Brandão et al., found no significant changes in radiotherapy for breast cancer patients [55] (Table 4).

According to Lan et al., the number of colorectal cancer patients who received chemotherapy was significantly higher in the MDTM cohort [23]. However, Ye et al., showed a significant decrease in overall cohort and stages I and IIA colorectal cancer, while stages IIB–IV showed no significant differences between the cohorts [35]. In lung cancer, Boxer et al., showed a significant increase of chemotherapy in the MDTM cohort [38], however Bydder et al., found no significant difference [39] (Table 4). In breast cancer, Brandão et al., showed that MDTM implementation did not lead to significant changes in the use of (neoadjuvant) chemotherapy or endocrine therapy [55].

Lan et al., showed a significant increase in the number of colorectal cancer patients in MDTM cohorts who received palliative chemotherapy, however MacDermid et al., showed no significant difference [23,24]. In lung cancer, two studies also showed a significant increase in palliative care [38,40], however three additional studies showed no significant difference between MDTM cohorts and control groups [39,43] (Table 4).

Finally, Tsai et al., found no significant differences between MDTM and control groups in any treatment combinations for breast cancer patients [56].

Muthukrishnan et al., provided no statistical analysis on these specific outcomes [47].

#### 3.4.4. Patient Outcomes

Eighteen studies reported on patient outcomes, i.e., survival, recurrence or metastasis, mortality and other patient-related outcomes (Table 5). These outcomes were not measured for prostate cancer.

Six out of eight studies that reported on survival showed a significant increase in survival of colorectal cancer patients that were discussed in MDTMs [18,23,24,25,35,37]. Of which, three studies only showed a significant effect of MDTMs on specific subgroups [24,25,37]. MacDermid et al., showed a significant increase in survival of patients with Dukes C disease while remaining unchanged for patients with Dukes B disease [24]. Similarly, Munro et al., showed that 5-year cause-specific survival for advanced cases increased significantly with MDTMs, while it did not change for early disease cases [25]. Foucan et al., showed a significantly increased survival duration after diagnosis, but not after surgery [37]. One out of eight studies reporting on survival did not show significant effects of MDTMs in colorectal cancer pathways [34], while one other study did not perform statistical analysis on the survival outcomes [27].

Besides survival, other patient-related outcomes were measured. Swellengrebel et al., showed no significant effect on resection margin rates [31]. Wille-Jørgensen et al. and Lan et al., showed that postoperative mortality decreased significantly after implementation of MDTMs in colorectal cancer [23,34], while recurrence and metastasis rates were not significantly affected [34]. Ye et al., showed a significantly lower tumor recurrence and longer time to recurrence for colorectal cancer patients who were discussed during MDTMs [35]. Chen et al. and Munro et al., both identified a significantly lower hazard ratio (HR) of death in the MDTM groups [18,25].

All five studies reporting on survival of lung cancer patients showed a significant improvement in survival of patients discussed during MDTMs [39,41,43,44,46]. Pan et al., showed a significantly lower adjusted hazard ratio (HR) of death in patients with stage III and IV non-small cell lung cancer that were discussed in MDTMs [41]. In particular, Hung et al., showed a prolonged length of survival for stage III lung cancer patients [46]. Quality of surgery was measured by Tamburini et al. and showed no significant differences between non-MDTM and MDTM groups, while overall mortality was significantly decreased [44].

In total, three studies reported on patient outcomes for breast cancer, of which all found a higher overall survival rate in the MDTM group compared to the control group [55,56,57]. Brandão et al., found a significantly higher survival rate in early breast cancer (Stage 0–III), while MDTM discussion in patients with metastatic breast cancer did not lead to a survival benefit. Furthermore, no significant increases in the proportion of clean surgical margins and complete axillary surgery were identified [55]. Brandão et al., found no significant differences in recurrence rate, while Tsai et al., showed a significant decrease in the MDTM group [55,56]. Yang et al., showed that survival was significantly higher in patients compliant with MDTM recommendations compared to the non-compliant group [57].

## 4. Discussion

### 4.1. Summary of Evidence

We systematically reviewed scientific literature and identified the impact MDTMs can have on colorectal, lung, prostate and breast cancer care. Overall, results showed that the implementation of MDTMs can have a significant impact on treatment decisions, patient outcomes and process outcomes (Table 6). However, all studies showed a low to very low quality of evidence and a critical or serious risk of bias. While our review suggests benefits of MDTMs in colorectal, lung, prostate and breast cancer care, there is need for more high-quality research.

Studies reporting on process outcomes such as cost- or time-related components are limited. Results suggest that MDTMs can have an effect on these process outcomes, however due to the limited evidence, no solid conclusions can be drawn. The systematic review by Ke et al., suggested that the investments in MDTMs are justified, but similar to our findings, Ke et al., also stated that there is a need for more rigorous studies on cost-effectiveness [58]. Some of the studies suggested that effects of MDTMs on management are less in early-stage (or non-recurrent) disease, which will typically constitute cases where the recommended treatment is more standardized and benefits of MDTM discussion may be limited [21,24,37,49,52]. This suggests a focus of future research on the cost-effectiveness of specific patient subgroups, i.e., early and advanced disease.

Results indicated that the impact on changes in management plans are different per cancer type and per hospital type (e.g., general hospital, university hospital). Overall, MDTM discussion resulted in more changes in lung cancer management plans (53–58%), compared to colorectal (6–29%), prostate (1.6–43%) and breast cancer (42.1%). This might be explained by the following aspects of lung cancer care. Typically, guidelines for lung cancer are less comprehensive and more frequently updated compared to, for example, prostate and breast cancer. In addition, management plans must be comprised in a short timespan due to the high mortality of lung cancer [59]. Of the 14 studies that reported on changes in management plans, 5 were performed in a university hospital, 5 were performed a tertiary referral cancer center and 4 in a general hospital. In all cancer types, general hospitals had the lowest percentage of changes in management plans, suggesting less impact of MDTMs on management plans in general hospitals. The latter might be the result of different case mix between general hospitals and tertiary referral centers or university hospitals. In addition, clinical trials might offer more diagnostic and therapy opportunities to be considered in university hospitals. Furthermore, due to the researchers’ affiliations with teaching and academic hospitals, more research is conducted there instead of general hospitals [1].

Studies that reported on changes in management plans generally showed a high risk of bias in measurement of outcomes, for several reasons. First, physicians who formulated the management plans prior to MDTMs often attended the meeting, and potentially affected the final recommendation with their opinion. Second, the final MDTM recommendation might also be influenced by knowledge of the initial plans, even when the physician who developed the initial plan did not attend the MDTM. Third, in some studies, the physicians who formulated the initial MDTM plans were also the outcome assessors that subjectively evaluated the changes made during the MDTM. A few studies minimized this outcome bias by blinding the MDTM to the initial plans or had an independent physician draw up the initial plans [29,45].

Studies that compared MDTM groups to (historical) control groups in terms of the number of patients that received certain types of diagnostics and/or treatment were focused on colorectal, lung and breast cancer patients. These outcomes were not reported for prostate cancer. All papers reporting on number of MRI scans and most papers on CT scans showed a significant increase in the MDTM cohort, suggesting more accurate staging. The systematic review of Pillay et al., showed similar outcomes, and also concluded that patients discussed at an MDTM were more likely to receive appropriate staging [7]. The results suggest that MDTM discussion often affects the treatment that patients received, typically less so for surgery, and different surgical types. Radiotherapy, chemotherapy and palliative care were chosen equally or more often. However, it is uncertain whether these trends are completely the result of MDTM discussion. Studies that reported on the impact of MDTMs on treatment patients received often compared the MDTM cohort with a control group over a long time period. For example, Lan et al., measured from 2001 to 2010, while MDTM was introduced in 2007. Therefore, differences in MDTM and control groups might also be affected by changes in techniques, guidelines and clinical practice. Lan et al., stated that there were significant differences in many aspects of the diagnosis and treatment during their measurement time, i.e., the introduction of targeted therapy. Thus, the impact of MDTMs might be overestimated.

Most studies investigating the effects of MDTMs on survival showed a significantly improved survival rate for colorectal, lung and breast cancer patients. A few studies identified the effects of MDTMs in certain patient subgroups based on disease stage and/or treatment combination [18,24,25,41,55,57]. Often, the overall survival rate was significantly improved, while some subgroups did not show significant differences in survival. Similar to the improved staging mentioned earlier, the improved survival might be partially explained by improvements in techniques, guidelines and clinical practice during the long measurement time of the studies.

Overall, most of the results in this systematic review are in agreement with previously published work. Results showed evidence for improved survival for colorectal, lung and breast cancer. Changes in clinical diagnostic and treatment decision making for colorectal, prostate and breast cancer was identified but rated as weak [7,8,60,61,62]. The reported patient and process outcomes in this systematic review were investigated in colorectal, lung and breast cancer, and not in prostate cancer. The review of Holmes et al., also concluded that the number of articles that studied the effect of MDTMs in prostate cancer is limited. Similar to our findings, Holmes et al., did encounter many abstracts, suggesting potential future publications on the topic [63]. For breast cancer, Blackwood et al., also showed evidence that an MDT approach is associated with improved clinical outcomes, however they did not report on the effect of MDTMs in particular [64].

### 4.2. Strengths and Limitations

Our systematic review has several strengths. Two independent researchers screened over 13,000 articles in a title–abstract screening and 165 articles in full-text, following the PRISMA-P 2015 and the Cochrane Collaboration’s double-date collection and extraction methodology [9,10]. We are the first to specifically evaluate and compare the effect of MDTMs on colorectal, lung, prostate and breast cancer care.

Our review also has some limitations. First, we acknowledge that selective reporting and publication bias cannot be ruled out. During the screening process, 27 abstracts met the inclusion criteria but were excluded due to a lack of information. In most cases, these abstracts did not result in a published paper, which might indicate a publication bias. Second, during the screening process, MDTMs might be misclassified as ‘multidisciplinary (MDT) approach’ due to limited or unclear information and the lack of a consistent definition for multidisciplinary team meetings [61,65]. Articles reporting the effect of ‘MDT approach’ were excluded, because ‘MDT approach’ is a broad term for collaboration between medical specialists, that may or may not include MDTMs. Subsequently, we cannot rule out exclusion of misclassified MDTMs. Third, general observations on MDTMs are limited due to small numbers of articles, inadequate statistical analysis and heterogeneity of patient and process outcomes. Finally, all studies included in this review had an observational study design. According to the GRADE scale, observational studies without special strengths or important limitations provide low quality of evidence. Therefore, the level of evidence of the included studies had to be rated low to very low. In contrast, an RCT without important limitations provides high-quality evidence according to the GRADE scale. However, MDTMs are often considered mandatory, and denying patients an MDTM discussion might be considered unethical. Therefore, an RCT might not be feasible in the evaluation of the effect of MDTMs. In conclusion, before discarding the low level of evidence, the level of evidence might never be higher than this.

### 4.3. Future Research

An unequal distribution of cancer types was identified in the literature with regard to the effects of MDTMs on treatment decisions, patient outcomes and process outcomes. As a result, current evidence of the potential benefits of MDTMs differs per cancer type, yet none of the evidence presented in the included studies is strong. In particular, current literature lacks studies that reported on the effect of MDTMs on patient outcomes and process outcomes for prostate cancer. Overall, high-quality research is required for all cancer types to confirm the potential benefit of MDTMs, preferably in a multicenter study with appropriate statistical analysis (e.g., a power calculation).

Another valuable focus for future research might be to investigate whether all patients should be discussed at MDTMs. Most studies showed that the majority of management plans did not change after MDTM discussion. Several of the included studies showed that MDTMs only had a significant effect on a specific subset of cancer patients, typically advanced cases. MDTMs are considered time-consuming and expensive, however these statements are mostly based on physicians/clinical experience. The increasing incidence of colorectal, lung, prostate and breast cancer might further pressure effort, time and financial resources. In this context, it might be important to re-evaluate the recommendations to discuss every patient in MDTMs and to focus on the cost-effectiveness of MDTMs. Cost-effectiveness studies on all four cancer types could be beneficial. In order to better understand the impact of MDTMs in addition to the clinical outcomes, cost-effectiveness studies would be essential, allowing for a critical evaluation of the effectiveness of MDTMs, based on clinical and process outcomes.

## 5. Conclusions

The number of studies that evaluated the effect of MDTMs is sparse, especially for lung, prostate and breast cancer compared to colorectal cancer. The reported evidence suggests that the implementation of MDTMs can have a significant impact on treatment decisions for colorectal, lung, prostate and breast cancer. In colorectal cancer, there is weak evidence that MDTMs result in more accurate staging. There is weak evidence that MDTMs improve patient outcomes (e.g., survival) for colorectal, lung and breast cancer patients.

## Figures and Tables

**Figure 1 cancers-13-04159-f001:**
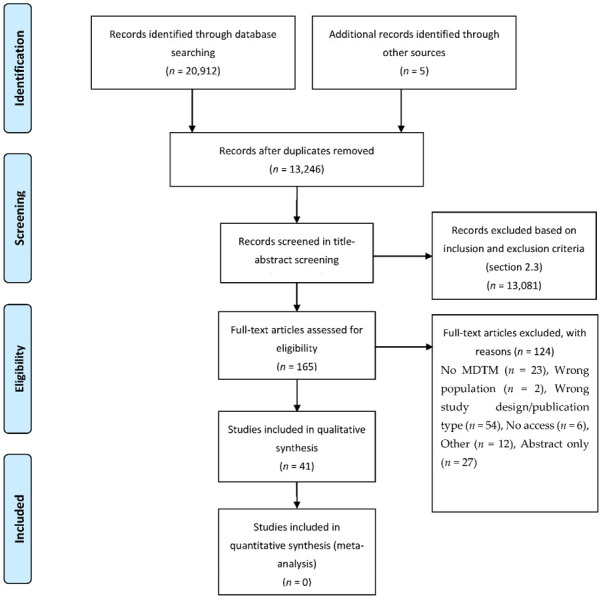
PRISMA-P 2015 flowchart.

**Table 1 cancers-13-04159-t001:** Study characteristics.

Author & Year	Cancer Type	Country	Aim	Hospital Type	Study Design	Inclusion Period	Participants
Acher et al., 2005 [48]	Urological cancers (prostate cancer)	UK	To examine the impact of MDTM on changes in management.	William Harvey Hospital (General hospital)	Prospective cohort study	6 months (date not specified)	124 discussions
Anania et al., 2019 [17]	Colorectal cancer (rectal cancer)	Italy	To compare the multidisciplinary experience group with the previous approach before the advent of the multidisciplinary program.	S. Anna Hospital (University hospital)	Retrospective before–after study	February 2007–April 2017 MDTM implementation in 2012	Non-MDTM: 45 patients MDTM: 51 patients Total: 96 patients
Boxer et al., 2011 [38]	Lung cancer	Australia	To evaluate the impact of MDTMs by comparing patterns of care among patients who were presented at an MDTM with those who were not presented at a meeting during the same period.	The Liverpool and Macarthur Cancer therapy centers (General hospital)	Prospective case-control study	1 December 2005–31 December 2008	Non-MDTM: 484 patients MDTM: 504 patients Total: 988 patients
Brandão et al., 2020 [55]	Breast cancer	Mozambique	To assess the impact of implementing an MDTM on the cost-effectiveness, care and survival.	Maputo Central hospital (General Hospitals)	Prospective before–after study	January 2015 and August 2017 Follow-up until November 2019 MDTM implementation: March 2016	Non-MDTM: 98 patients MDTM: 107 patients Total: 205 patients
Bydder et al., 2009 [39]	Non-small cell lung cancer	Australia	To examine the proportion of patients that is discussed by the MDTM and the impact on the treatment and survival.	Sir Charles Gairdner Hospital (Tertiary Hospitals)	Prospective case-control study	2006 Follow-up until 31 March 2008	Non-MDTM: 17 patients MDTM: 81 patients Total: 98 patients
Chen et al., 2018 [18]	Colorectal cancer with lung or liver metastasis	Taiwan	To investigate whether MDTM intervention is associated with improved survival.	Wan Fang Hospital (University hospital)	Case-control study	January 2007–December 2017 Mean follow-up: 84 months ± 35 months	Non-MDTM: 86 patients MDTM: 75 patients Total: 161 patients
Chinai et al., 2013	Colorectal cancer	UK	To evaluate the clinical impact and cost-effectiveness of a MDTM.	Derriford Hospital Plymouth Hospitals (General hospitals)	Prospective cohort study	3 months (date not specified)	47 patients
De Luca et al., 2019 [49]	Prostate cancer	Italy	To investigate the impact on clinical management of the uro-oncology MDTM.	San Luigi Hospital (Tertiary hospital)	Prospective cohort study	Jan 2016–June 2017	201 patients/272 discussions
El Khoury et al., 2016 [50]	Urological cancers (prostate cancer)	Lebanon	To examine the impact of MDTMs on the management decision for urological cancers.	Notre-Dame de Secours University Medical Centre (University hospital)	Prospective cohort study	July 2012–July 2014.	Prostate: 82 patients
Fernando et al., 2017 [20]	Colorectal cancer	New Zealand	The primary objective to determine which patients benefit most from MDTMs, and secondarily to determine whether there was a group of patients which could be managed by protocol without discussion at the MDTM.	Christchurch Hospital (Tertiary hospital)	Prospective case-control study	1 September 2013–1 November 2014	Non-MDTM: 182 patients MDTM: 459 patients Total: 641 patients
Foucan et al., 2020 [37]	Colorectal cancer (colon cancer)	France	To evaluate the factors associated with the non-presentation in MDTM, and to assess the association between non-MDTM and therapeutic care management.	Multicenter	Retrospective case-control study	2010 (date not specified)	Non-MDTM: 142 patients MDTM: 431 patients Total: 573 patients
Freeman et al., 2015 [40]	Non-small cell lung cancer	US	To compare quality and cost metrics for propensity-matched MDTM patients to patients without access to such care coordination across a geographically diverse system of hospitals.	Multicenter	Retrospective case-control study	2008–2012 (date not specified)	Non-MDTM: 6627 patients MDTM: 6627 patients Total: 13,254 patients
Hung et al., 2020 [46]	Non-small cell lung cancer	Taiwan	To prove MDTM discussion could prolong the average time of survival for patients with stage III NSCLC.	Taipei Veterans General Hospital (General hospital)	Retrospective before–after study	January 2013–December 2018 MDTM implementation February 2016	Non-MDTM: 273 patients MDTM: 242 patients Total: 515 patients
Jung et al., 2018 [21]	Colorectal cancer	Korea	To assess the impact of MDTM on clinical decision making.	Asan Medical Center (Tertiary hospital)	Prospective cohort study	1 January 2011–31 December 2014	1383 patients
Karagkounis et al., 2018 [22]	Colorectal cancer (rectal cancer)	US	To determine the frequency and manner in which MDTM changed the management of patients.	Cleveland clinic (University hospital)	Prospective cohort study	July 2015–June 2016	316 patients/414 discussions
Kurpad et al., 2011 [51]	Genitourinary cancers (prostate cancer)	US	To study the effect of MDTMs on the diagnosis and treatment decisions of new patients.	Lineberger Comprehensive Cancer Center (University hospital)	Prospective cohort study	June 2007–June 2008	Prostate: 92 patients
Lan et al., 2016 [23]	Colorectal cancer	Taiwan	Analyzing and comparing the outcomes of colorectal cancer patients with metastatic disease before and after the era of MDTM.	Taipei Veterans General Hospital (Tertiary hospital)	Before–after study	January 2001–December 2010 MDTM implementation: October 2007	Non-MDTM: 636 patients MDTM: 439 patients Total: 1075 patients
MacDermid et al., 2009 [24]	Colorectal cancer	UK	To assess the effect of this on patient’s survival, and trends in the use of adjuvant chemotherapy.	Royal Alexandra Hospital (General hospital)	Before–after study	January 1997–December 2005 MDTM implementation: June 2002	Non-MDTM: 176 patients MDTM: 134 patients Total: 310 patients
Maurizi et al., 2017 [36]	Colorectal cancer (rectal cancer)	Italy	To evaluate the improvements on rectal cancer treatment outcomes after the introduction of the MDTMs.	Carlo Urbani hospital (General hospital)	Before–after study	January 2014–December 2015 MDTM implementation: January 2015	Non-MDTM: 30 patients MDTM: 35 patients Total: 65 patients
Munro et al., 2015 [25]	Colorectal cancer	UK	To review the effect of MDTM, and implementation of recommendations, on survival.	Hospitals in Tayside region in Eastern Scotland (General hospitals)	Case-control study	1 January 2006–31 December 2007 Mean follow-up: 73.3 months	Non-MDTM: 175 patients MDTM: 411 patients Total: 586 patients
Murthy et al., 2014 [54]	Breast	US	To investigate the role of MDTM on patient management and how this led to treatment modifications.	Saint Barnabas Medical Center (General hospital)	Prospective cohort study	June 2010–June 2011	242 patients
Muthukrishnan et al., 2020 [47]	Lung cancer	US	To investigate whether early MDTM discussions affected the time required to complete a lung cancer evaluation.	Metrohealth Medical Center (University hospital)	Retrospective case-control study	December 2015–January 2017	Non-MDTM: 106 patients MDTM: 55 patients Total: 161 patients
Nikolovski et al., 2017 [26]	Colorectal cancer	Australia	To determine whether the introduction of MDTM altered the length of time to treatment.	Geelong Hospital (University hospital)	Before–after study	1 January 2006–3 February 2011	Non-MDTM Historical control: 56 patients Non-MDTM: 259 patients MDTM: 82 patients Total: 397 patients
Palmer et al., 2011 [27]	Colorectal cancer (rectal cancer)	Sweden	To assess outcome in relation to preoperative local and distant staging, with or without MDTM.	Hospitals in Stockholm-Gotland region (University hospitals)	Prospective case-control study	1995–2004 Follow-up: March 2008	Non MDTM: 99 patients MDTM: 65 patients Total: 303 patients
Pan et al., 2015 [41]	Non-small cell lung cancer	Taiwan	To analyze the factors affecting survival, at each stage of NSCLC.	Multicenter	Retrospective cohort study	2005–2011	Non-MDTM: 27,937 patients MDTM: 4632 patients Total: 32,569 patients
Rao et al., 2014 [52]	Urological cancers (prostate cancer)	Australia	To analyze the impact of the uro-oncology MDTMs on patient management decisions, and to develop criteria for patient inclusion in MDTMs.	Austin Hospital (Tertiary hospital)	Prospective cohort study	3 month period in 2012 (date not specified)	Prostate: 47 discussions
Richardson et al., 2016 [28]	Colorectal cancer (rectal cancer)	US	To assess whether MDTM participation improves process evaluation, outcomes and technical aspects of surgery.	Baylor University Medical Center (University hospital)	Retrospective before–after study	2011–2014 MDTM implementation: January 2013	Non-MDTM: 42 patients MDTM (2013): 41 patients MDTM (2014): 47 patients Total: 130 patients
Ryan et al., 2014 [29]	Colorectal cancer	Australia	To evaluate prospectively the colorectal MDTM to determine the utility of the meeting.	Western Health Melbourne (General hospital)	Prospective cohort study	6 months (date not specified)	197 patients/261 discussions
Scarberry et al., 2018 [53]	Genitourinary cancers (prostate cancer)	US	To prospectively evaluate the effectiveness of MDTM on altering treatment plans.	University hospital Cleveland Medical Center (University hospital)	Prospective cohort study	September 2011–April 2013	Prostate cancer: 125 patients
Schmidt et al., 2015 [42]	Thoracic cancer (lung cancer)	US	To analyze the actual impact of MDTM presentation on decision making in thoracic cancer cases.	Virginia Mason Medical Center (General hospital)	Prospective cohort study	1 June 2010–31 December 2012	Lung cancer: 294 patients (451 discussions)
Snelgrove et al., 2015 [30]	Colorectal cancer (rectal cancer)	Canada	To assess: (1) the quality of MDTM, (2) the effect of MDTM on the initial treatment plan, (3) compliance with the MDTM treatment recommendation and (4) clinical outcomes.	Mount Sinai Hospital (University hospital)	Prospective cohort study	1 September 2012–30 June 2013	42 patients
Stone et al., 2018 [43]	Lung cancer	Australia	To evaluate outcomes including survival, according to MDTM presentation and to explore the utility of data obtained from local clinical sources.	St Vincent’s Hospital (University hospital)	Prospective case-control study	1 January 2006–31 December 2012 Follow-up: 23 May 2014.	Non-MDTM: 295 patients MDTM: 902 patients Total: 1197 patients
Swellengrebel et al., 2011 [31]	Colorectal cancer (rectal cancer)	The Netherlands	To evaluate the additional value of MDTM discussion, with the occurrence of a positive CRM as an endpoint.	Antoni van Leeuwenhoek Netherlands cancer Institute (Tertiary hospital)	Case-control study	January 2006–January 2008	Non- MDTM: 94 patients MDTM: 116 patients Total: 210 patients
Tamburini et al., 2018 [44]	Non-small cell lung cancer	Italy	To evaluate the impact of MDTM on survival of patients undergoing surgery for NSCLC.	Ferrara University Hospital (University hospital)	Before–after study	January 2008–December 2015 MDTM implementation: 2012	Non-MDTM: 246 patients MDTM: 186 patients Total: 432 patients
Tsai et al., 2020 [56]	Breast cancer	Taiwan	To investigate the influence of MDTM on the risk of recurrence and death.	Multicenter	Retrospective case-control study	2004–2010	Non-MDTM: 9266 patients MDTM: 9266 patients Total: 18,532 patients
Ung et al., 2016 [45]	Lung cancer	Australia	To measure the impact of MDTM on clinicians’ management plans, and the implementation rate of the meeting recommendations.	Peter MacCallum Cancer Centre (Tertiary referral center)	Prospective cohort study	March–May 2011	68 patients
Vaughan-Shaw et al., 2015 [32]	Colorectal cancer (rectal cancer)	UK	To assess the impact of the introduction of a specialist early rectal cancer MDTM on the investigation and management of rectal cancer.	Cheltenham General Hospital (General hospital)	Before–after study	24 months (2006 and 2011) MDTM implementation: 2011	Non-MDTM: 19 patients MDTM: 24 patients Total: 43 patients
Wanis et al., 2017 [33]	Colorectal cancer with liver metastasis	Canada	To determine the access to and association between MDTM review and management amongst patients with colorectal cancer and synchronous liver metastases.	London Health Sciences Centre (University Hospital)	Retrospective case-control study	January 2008–June 2015	Non-MDTM: 37 patients MDTM: 29 patients Total: 66 patients
Wille-Jørgensen et al., 2013 [34]	Colorectal cancer (rectal cancer)	Denmark	To compare the outcomes of patients before and after the establishment of MDTMs in the two surgical departments in Copenhagen.	Bispebjerg and Hvidovre Hospitals (University hospitals)	Before–after study	1 May 2001–31 August 2006 MDTM implementation: September 2004	Non-MDTM: 467 patients MDTM: 344 patients Total: 811 patients
Yang et al., 2020 [57]	Breast cancer	China	To identify which clinicopathological characteristics may influence compliance with MDTM recommendations, and to evaluate whether MDTM compliance affects the prognosis of early breast cancer.	Shanghai Ruijin Hospital (General hospital)	Retrospective cohort study	April 2013–August 2018 Mean follow-up: 32.75 months	4501 patients
Ye et al., 2012 [35]	Colorectal cancer	China	To assess the effect on management of colorectal cancer after the inception of an MDTM.	Peking University People’s Hospital (University hospital)	Before–after study	January 1999–September 2006 MDTM implementation: December 2002	Non-MDTM: 297 patients MDTM: 298 patients Total: 595 patients

MDTM = multidisciplinary team meeting, NSCLC = non-small cell lung cancer, UK = United Kingdom, US = United States. Study design definitions: Cohort study = comparisons in outcomes made within one cohort; Before–after study = comparison of control group (no-MDTM) and MDTM group before and after implementation of MDTMs; Case-control study = comparison of control group (no-MDTM) with MDTM group, within the same time period.

**Table 2 cancers-13-04159-t002:** Results of process outcomes.

Author and Year	Cancer Type	Outcomes	Study Results	
**Time Management**				
Boxer et al., 2011 [38]	Lung cancer	Time from diagnosis to treatment (days)	Surgery	50 vs. 42 days, *p* = 0.49
			Curative radiotherapy	91 vs. 106 days, *p* = 0.65
			Palliative radiotherapy	89 vs. 87 days, *p* = 0.89
			Curative chemotherapy	45 vs. 45 days, *p* = 0.97
			Palliative chemotherapy	44 vs. 60 days, ***p* = 0.03**
			Palliative care	100 vs. 110 days, *p* = 0.37
Brandão et al., 2020 [55]	Breast cancer	Time from diagnosis to treatment (% patients)	Less than 45 days	44.7% vs. 51.9%
			45 days or longer	55.3% vs. 48.1%
				*p* = 0.324
Foucan et al., 2020 [37]	Colorectal cancer (colon cancer)	Time from diagnosis to surgery (days)	All surgeries	21.7 vs. 34.6 days
			Emergency surgery	8.0 vs. 10.1 days
			Non-emergency surgery	26.2 vs. 38.7 days
Freeman et al., 2015 [40]	Non-small cell lung cancer	Time from diagnosis to treatment (days)	32 ± 11 vs. 19 ± 8 days, ***p* < 0.001**	
Muthukrishnan et al., 2020 [47]	Lung cancer	Time from diagnosis to treatment (days)	Imaging to staging: Total 49.33 vs. 70.15 days ***p* < 0.001**	
			Stage I–III 65.45 vs. 75.77 days *p* = 0.39	
			Imaging to diagnosis: Total 37.36 vs. 61.71 days ***p* < 0.001**	
			Stage I–III 54.31 vs. 69.71 days *p* = 0.13	
			Diagnosis to staging: Total 11.97 vs. 8.44 days *p* = 0.07	
			Stage I–III 11.14 vs. 6.06 days *p* = 0.07	
			Staging to therapy: Total 25.23 vs. 44.69 days ***p* = 0.01**	
			Stage I–III 24.24 vs. 41.46 days ***p* = 0.03**	
			Diagnosis to therapy: Total 37.20 vs. 53.13 days *p* = 0.06	
			Stage I–III 35.37 vs. 47.51 days *p* = 0.28	
			Imaging to therapy: Total 74.56 vs. 114.84 days ***p* < 0.001**	
			Stage I–III 89.69 vs. 117.23 days *p* = 0.15	
Nikolovski et al., 2017 [26]	Colorectal cancer	Time from diagnosis to treatment (days) Time from diagnosis to surgery (days)	Historical control vs. MDTM group	Concurrent control vs. MDTM group
			Total: 19.5 vs. 30 days, ***p* = 0.001**	Total: 18 vs. 30 days, ***p* < 0.001**
			Colon: 14 vs. 18 days, *p* = 0.338	Colon: 15 vs. 18 days, *p* = 0.348
			Rectal: 25 vs. 32.5 days, *p* = 0.090	Rectal: 23 vs. 32.5 days, ***p* < 0.001**
			Total: 17 vs. 22 days, *p* = 0.061	Total: 17 vs. 22 days, ***p* = 0.002**
			Colon: 14 vs. 18 days, *p* = 0.406	Colon: 15 vs. 18 days, *p* = 0.384
			Rectal: 21 vs. 24 days, *p* = 0.367	Rectal: 21 vs. 24 days, *p* = 0.085
Stone et al., 2018 [43]	Lung cancer	Time to referral to palliative care for stage IV patients (days)	26 vs. 69 days	
**Costs**				
Brandão et al., 2020 [55]	Breast cancer	Cost-effectiveness (USD $)	3-year cost increase of implementing MDTM: $119.83 per patient. Incremental cost-effectiveness ratio: $802.96 per QALY. MDTM implementation is a cost-effective measure.	
Chinai et al., 2013 [19]	Colorectal cancer	Estimated costs (£)	Estimated annual costs of MDTM: £162,734	
Freeman et al., 2015 [40]	Non-small cell lung cancer	Mean cost of care (USD $)	$10,213 vs. $7,212; ***p* < 0.001**	
**Other**				
Freeman et al., 2015 [40]	Non-small cell lung cancer	(1) Research participation offered (2) Adherence to NCCN guidelines	(1) 6% vs. 17%, ***p* < 0.001** (2) 71% vs. 88%, ***p* < 0.001**	
Ryan et al., 2014 [29]	Colorectal cancer	Benefit of MDTM discussion	Discussions were considered beneficial in 26.8% * of all discussions.	

Results, if available, comparing patients in MDTM group and non-MDTM group, are shown as (‘results non-MDTM group’ vs. ‘results MDTM group’, *p*-value). * = These values were calculated by the authors (L.K. and H.W.). MDTM = multidisciplinary team meeting, TME = total mesorectal excision. P-values < 0.05 were considered significant, annotated in bold.

**Table 3 cancers-13-04159-t003:** Results for changes in management plans.

Author and Year	Cancer Type	Proportion of Cases with Changed Overall Management Plans	Proportion of Changed Cases Stratified per Stage or MDTM Type	Type of Changes
Acher et al., 2005 [48]	Urological cancers (prostate cancer)	1.6% *	-	-
Chinai et al., 2013 [19]	Colorectal cancer	6.4%	-	-
De Luca et al., 2019 [49]	Prostate cancer	35.8% *	Local disease: 23.2%, advanced disease: 46.9%, metastatic disease: 33.4%	-
El Khoury et al., 2016 [50]	Urological cancers (prostate cancer)	42.7%	Gleason score 6: 27.3% Gleason score 7: 51.7% Gleason score 8: 44.4% Gleason score 9: 50.0% Gleason score 10: 40.0%	-
Fernando et al., 2017 [20]	Colorectal cancer	23%	-	Proportion of changed clinical staging cases: 4%
Jung et al., 2018 [21]	Colorectal cancer	12.9%	Newly diagnosed cancer: 7.6%, recurrence cancer patients: 16.4% (*p* < 0.001)	Treatment plans overall: 12.9% Nonsurgical treatment in 66.5% of cases, modifications to the surgical approach: 3.4% no treatment: 30.2%.
Karagkounis et al., 2018 [22]	Rectal cancer	26.1%	Initial discussion: 32.4%, Follow-up discussions: 35.2%, Postoperative discussions: 6.3% (*p* < 0.001)	Diagnostic plan: 9.7% * of cases Treatment plans: 20.5% * Decided to operate: 2.1% * of cases; Decided not to operate: 12.4% *; Changed operative approach: 18.6% *. Neoadjuvant therapy added: 39.2% *; Adjuvant therapy added: 6.2% *. Additional workup biopsy/pathology: 10.3% *; Additional workup imaging: 26.8% *. Changes were more frequent when the pre-MDTM plan was considered tentative by the attending physician (45.5%, *p* < 0.001).
Kurpad et al., 2011 [51]	Urological cancers (prostate cancer)	50%	-	Treatment changed in 18.5%, diagnosis changed in 6.5%, both diagnosis and treatment changed in 3.3%, other changes in 7.6% and N/A in 14.1% of cases.
Murthy et al., 2014 [54]	Breast cancer	42%	Stage 0: 21%, stage IA: 27%, stage IB: 8%, stage IIA: 15%, stage IIB: 17%, stage IIIA: 7%, stage IIIB: 0%, stage IIIC: 1%, stage IV: 2%.	Surgical treatment: 38.2% of all changes. medical management (chemotherapy/endocrine therapy): 33.3% Radiation treatment: 16.6% Combined medical and radiation therapy: 6.8% Imaging changes (e.g., MRI, mammogram): 4.9%
Rao et al., 2014 [52]	Urological cancers (prostate cancer)	26%	T1: 0%, T2: 25%, T3: 21% (*p* = 0.62). Localized disease: 23% Metastatic disease: 38% (*p* < 0.05).	-
Scarberry et al., 2018 [53]	Genitourinary cancers (Prostate)	17.6%	-	-
Schmidt et al., 2015 [42]	Lung cancer	53%	-	Changes in treatment: 41% of cases Staging recommendations changed: 59% of cases
Snelgrove et al., 2015 [30]	Rectal cancer	29%	-	Changes in initial treatment: Primary surgery 58% * Neoadjuvant chemoradiation: 25% * Systemic chemotherapy: 16.7% *
Ung et al., 2016 [45]	Lung cancer	58%	-	Additional investigations: 59% Treatment modality: 19% Treatment intent: 9% Tumor histology: 6% Tumor stage: 6%

* These values were calculated by the authors (L.K. and H.W.). Results, if available, comparing patients in MDTM group and non-MDTM group, are shown as (‘results non-MDTM group’ vs. ‘results MDTM group’, *p*-value). MDTM = multidisciplinary team meeting.

**Table 4 cancers-13-04159-t004:** Results for diagnostics, treatment and palliative care.

Author and Year	Cancer Type	Outcomes	Study Results (Proportion of Patients Received)
**Imaging/Staging/Diagnostics**			
Anania et al., 2019 [17]	Colorectal cancer	(1) Colonoscopy (2) CT (3) MRI (4) Ultrasound	(1) 57.7% * vs 78.4% * (2) 24.4% vs. 82.4%; ***p* < 0.01** (3) 4.4% vs. 62.7%; ***p* < 0.01** (4) 15.6% vs. 23.5%; *p* = NS
Fernando et al., 2017 [20]	Colorectal cancer	(1) CT abdomen (2) CT chest (3) Colonography (4) MRI (5) FDG-PET CT (6) X-ray chest (7) Ultrasound (8) col/sigmoidoscopy (9) biopsy (10) liver function tests (11) Carcinoembryonic antigen	(1) 96.2% vs. 96.3%; *p* = 0.545 (2) 23.6% vs. 49.5%; ***p* < 0.001** (3) 25.8% vs. 19.6%; *p* = 0.054 (4) 9.3% vs. 67.5%; ***p* < 0.001** (5) 2.2% vs. 17.9%; ***p* < 0.001** (6) 75.3% vs. 43.4% ***p* < 0.001** (7) 5.5% vs. 6.1%; *p* = 0.467 (8) 85.7% vs. 89.3%; *p* = 0.127 (9) 81.9% vs. 88.2%; ***p* = 0.025** (10) 23.6% vs. 29.8%; *p* = 0.068 (11) 79.1% vs. 78.2%; *p* = 0.446
Foucan et al., 2020 [37]	Colon cancer	CT scans	46.6% vs. 66.6%; ***p* < 0.001**
Freeman et al., 2015 [40]	NSCLC	Complete staging	67% vs. 91%; ***p* < 0.001**
Maurizi et al., 2017 [36]	Rectal cancer	(1) MRI, (2) CEA testing (3) colonoscopy (4) CT (5) Endoscopic rectal ultrasound (6) Post-therapy preoperative restaging with MRI	(1) 23.33% vs. 51.43%; ***p* = 0.010** (2) 46.67% vs. 65.71%; *p* = 0.061 (3) 86.67% vs. 85.71; *p* = 0.456 (4) 90.00% vs. 97.14%; *p* = 0.116 (5) 16.67% vs. 25.71%; *p* = 0.188 (6) 26.67% vs. 34.29%; *p* = 0.254
Richardson et al., 2016 [28]	Rectal cancer	(1) Colonoscopy	(1) 95% vs. 100% (MDTM year 1) vs. 96% (MDTM year (2) *p* = 0.3828
Palmer et al., 2011 [27]	Rectal cancer	Preoperative local staging: (1) MRI exams (2) Endorectal ultrasonography Preoperative distant staging (3) CT/MRI abdomen (4) Ultrasound abdomen (5) Chest CT or X-ray	(1) 89.9% * vs. 98.5% * (2) 21.2% * vs 4.6% * (3) 57.6% * vs. 75.4% * (4) 52.5% * vs 40.0% * (5) 100% * vs 100%*
Tamburini et al., 2018 [44]	NSCLC	Rate of complete preoperative evaluation	64% vs. 93% (***p* < 0.001**).
Ye at al. 2012 [35]	Colorectal cancer	(1) CT examination performed before operation (2) CT TNM staging performed before operation (3) accurate TNM staging	(1) 30.3% vs. 55.7%; ***p* < 0.001** (2) 41.1% vs. 81.3%; ***p* < 0.001** (3) 45.9% vs. 64.0%; ***p* = 0.044**
**Surgery**			
Anania et al., 2019 [17]	Colorectal cancer	Surgical type: (1) Total mesorectal excision (2) Laparoscopic total mesorectal excision (3) Open miles procedure (4) Laparoscopic miles procedure	(1) 2.2% vs. 13.7% (2) 88.9% vs. 68.6% (3) 2.2% vs. 0% (4) 6.7% vs 17.6
Boxer et al., 2011 [38]	Lung cancer	(1) Surgery (2) Surgery stratified per stage	(1) 13% vs. 12%; *p* = 0.84 (2) NSCLC stages I +II (61% vs. 49%; *p* = 0.25) NSCLC stage III (26% vs. 16%; *p* = 0.16) NSCLC stage IV (0% vs. 2%; *p* = 0.13)
Brandão et al., 2020 [55]	Breast cancer	(1) Surgery (ever) (2) Surgery type (first treatment) (a) Total mastectomy (b) Tumorectomy	(1) 80.6% vs. 82.2% (*p* = 0.858) (2) (a) 94.9% vs. 89.8% (b) 5.1% vs. 10.2 *p* = 0.257
Foucan et al., 2020 [37]	Colon cancer	Surgery	61.3% vs. 86.8%; ***p* = 0.004**
Freeman et al., 2015 [40]	NSCLC	Non-therapeutic surgical procedure	4% vs. 2%; ***p* < 0.001**
Lan et al., 2016 [23]	Colorectal cancer	Surgical resection of: (1) primary tumor (2) metastatic foci (3) liver metastasis (4) lung metastasis	(1) 88.5% vs 82.7%; ***p* = 0.007** (2) 21.7% vs. 29.8%; ***p* = 0.003** (3) 19.6% vs. 35.2%; ***p* < 0.001** (4) 12.4% vs. 14.3%; *p* = 0.803
Muthukrishnan et al., 2020 [47]	Lung cancer	Surgery	17.0% vs. 16.4%
Palmer et al.* 2011 [27]	Rectal cancer	Types of surgery: (1) no surgery (2) explorative laparotomy (3) resection of rectal cancer	(1) 4.0% vs. 7.7% (2) 21.1% vs. 10.8% (3) 74.7% vs. 81.5% ***p*** **= 0.024**
Richardson et al., 2016 [28]	Rectal cancer	(1) Appropriate abdomino-peritoneal resections treatment (2) Surgical type: (a) Transanal (minimally) invasive surgery (b) Low Anterior Resection (c) Transanal Transabdominal Low Anterior Resection (d) Abdominal-peritoneal resection	(1) 50% vs. 71% (MDTM year 1) vs. 78% (MDTM year 2); *p* = 0.191 (2) (a) 22% vs. 18% (MDTM year 1) vs. 9% (MDTM year 2) (b) 69% vs. 63% (MDTM year 1) vs. 40% (MDTM year 2) (c) 0% vs. 3% (MDTM year 1) vs. 7% (MDTM year 2) (d) 10% vs. 18% (MDTM year 1) vs. 43% (MDTM year 2); *p* = 0.002
Swellengrebel et al., 2011 [31]	Rectal cancer	(1) Surgical type: (a) Low Anterior Resection (b) Hartmann procedure (c) Abdominal Perineal Resection (d) No surgery	(1) (a) 71% vs 41% (b) 9% vs 17% (c) 19% vs 40% (d) 1% vs 2%
Tamburini et al., 2018 [44]	NSCLC	(1) Surgical type: (a) video-assisted thoracoscopic surgery (b) exploratory thoracotomy	(1) (a) 48% vs. 9%; ***p* = 0.001** (b) 3% vs 1.8%; *p* = 0.31
Vaughan-Shaw et al., 2015 [32]	Rectal cancer	Surgical treatment: (1) local excision (2) less resection (3) declined surgery	(1) 15.8% vs 83.3% (2) 79.0% vs. 16.7% (3) 5.3% vs. 0%
Wanis et al., 2017 [33]	Colorectal cancer with liver metastasis	Resection order	MDTM group significantly more likely (***p* < 0.001**) to undergo simultaneous resection of the primary colorectal tumor and liver metastases.
Wille-Jørgensen et al., 2013 [34]	Rectal cancer	Surgery	88% vs. 86%
**Radiotherapy and Chemotherapy**			
Boxer et al., 2011 [38]	Lung cancer	(1) Radiotherapy (a) overall (b) stratified per NSCLC stage (c) stratified per SCLC stage (2) Chemotherapy (a) overall (b) stratified per NSCLC stage (c) stratified per SCLC stage	(1) (a) 33% vs. 66%; ***p* < 0.001**. (b) stages I + II (17% vs. 54%; ***p* < 0.001**) stage III (46% vs. 71%; ***p* = 0.01**) stage IV (43% vs. 68%; ***p* < 0.001**) (c) limited stage (71% vs. 89%; *p* = 0.28), extensive stage (46% vs. 50%; *p* = 0.72). (2) (a) 29% vs. 46%; ***p* = 0.001**. (b) stages I + II (15% vs. 18%; *p* = 0.67), stage III (39% vs. 43%; *p* = 0.72), stage IV (29% vs. 42%; ***p* = 0.01**), (c) SCLC limited stage (71% vs. 100%; *p* = 0.72), SCLC extensive stage (76% vs. 75%; *p* = 0.89).
Brandão et al., 2020 [55]	Breast cancer	(1) Radiotherapy (first treatment) (2) Chemotherapy	(1) *p* = 0.175 (2) 91.8% vs. 96.3%; *p* = 0.237
Bydder et al., 2009 [39]	NSCLC	(1) Radical radiotherapy/Chemoradiotherapy (2) Chemotherapy	(1) 6% vs. 10%; *p* = 0.318 (2) 29% vs. 42%; *p* = 0.141
Freeman et al., 2015 [40]	NSCLC	Radio and/or chemotherapy before tissue diagnosis	5% vs. 3%; ***p* < 0.001**
Lan et al., 2016 [23]	Colorectal cancer	(1) Chemotherapy (2) Radiotherapy	(1) 75.9% vs. 83.8 %; ***p* = 0.002** (2) 9.6% vs. 20.5 %; ***p* < 0.001**
MacDermid et al., 2009 [24]	Colorectal cancer	(1) Primary adjuvant chemotherapy (a) Overall (b) Dukes B (c) Dukes C (2) Preoperative radiotherapy	(1) (a) 13% vs. 31.3%; ***p* < 0.001** (b)1.5% vs. 17.6%; ***p* = 0.002** (c) 31.9% vs. 58.6%; ***p* = 0.004**(2) 24.4% vs. 32.5%; *p* = 0.462
Muthukrishnan et al., 2020 [47]	Lung cancer	(1) Radiation (2) Chemotherapy (3) Chemo-radiation.	(1) 17.9% vs. 20.0% (2) 14.2% vs. 18.2% (3) 28.3% vs. 32.7%
Palmer et al., * 2011 [27]	Rectal cancer	Preoperative treatments: (1) No treatment (2) Short radiotherapy (3) Long radiotherapy (4) Radio-chemotherapy (5) Chemotherapy (6) Unknown	(1) 42.4% vs. 21.5% (2) 41.4% vs. 13.8% (3) 5.1% vs. 30.8% (4) 9.1% vs. 29.2% (5) 1.0% vs 1.5% (6) 1.0% vs. 3.1%
Wanis et al., 2017 [33]	Colorectal cancer with liver metastasis	Chemotherapy	NS
Wille-Jørgensen et al., 2013 [34]	Rectal cancer	Preoperative (chemo)radiotherapy	19% vs. 25%
Ye et al., 2012 [35]	Colorectal cancer	(1) Adjuvant chemotherapy (a) Overall (b) stage I (c) stage IIA (d) stage IIB (e) stage IIIA (f) stage IIIB (g) stage IIIC (h) stage IV (2) Adjuvant radiotherapy	(1) (a) 82.8%* vs. 49.3% *; ***p* < 0.001** (b) 64.4%* vs 0% *; ***p* < 0.001** (c) 82.2%* vs. 12.2% *; ***p* < 0.001** (d) 80.0%* vs. 100% *; *p* = 0.183 (e) 84.6%* vs. 90.9%; *p* = 0.577 (f) 93.0% * vs. 91.1% *; *p* = 0.728 (g) 93.1% * vs. 86.1% *; *p* = 616 (h) 83.3% * vs. 88.4% *; *p* = 0.750 2) 0.3% * vs. 10.1% *; ***p* < 0.001**
**Palliative Care and Hospice Referral**			
Boxer et al., 2011 [38]	Lung cancer	Referral to palliative care	53% vs 66%; ***p* < 0.001**
Bydder et al., 2009 [39]	NSCLC	(1) Palliative radiotherapy only (2) Palliative care only	(1) 35% vs. 25%; *p* = 0.152 (2) 29% vs. 23%; *p* = 0.204
Freeman et al., 2015 [40]	NSCLC	Palliative or hospice care	4% vs. 9%; ***p* < 0.001**
MacDermid et al., 2009 [24]	Colorectal cancer	Palliative chemotherapy	32.5% vs. 44%; *p* = 0.431
Muthukrishnan et al., 2020 [47]	Lung cancer	Hospice referral	22.6% vs. 12.7%
Stone et al., 2018 [43]	Lung cancer	Referral to palliative care	78.0% vs. 85.3%; *p* = 0.06
**Other**			
Anania et al., 2019 [17]	Colorectal cancer	Neo-adjuvant therapy	The MDTM cohort showed a significantly higher use of neo-adjuvant therapy (22.2% vs. 56.9%; ***p* < 0.01**).
Brandão et al., 2020 [55]	Breast cancer	Endocrine therapy (first treatment)	*p* = 0.888
Bydder et al., 2009 [39]	NSCLC	‘Active’ treatment	35% vs. 52%; *p* = 0.288
Foucan et al., 2020 [37]	Colon cancer	Treatment type (surgery and chemotherapy): (1) Overall (2) Stratified per stage: (a) Stage I (b) Stage II (c) Stage III (d) Stage IV	(1) NS (2) (a) *p* = 1.00 (b) 0.869 (c) *p* = 0.042 (d) ***p* < 0.001**
Maurizi et al., 2017 [36]	Rectal cancer	Neoadjuvant therapy	33.33% vs. 42.86%; *p* = 0.216
Tsai et al., 2020 [56]	Breast cancer	Treatment combinations	*p* = 0.211

Results, if available, comparing patients in MDTM group and non-MDTM group, are shown as (‘results non-MDTM group’ vs. ‘results MDTM group’, *p*-value). MDTM = multidisciplinary team meeting, NSCLC = non-small cell lung cancer SCLC = small cell lung cancer. * = percentages where calculated by the authors (L.K. and H.W.). P-values < 0.05 were considered significant, annotated in bold.

**Table 5 cancers-13-04159-t005:** Results of patient outcomes.

**Author and Year**	**Cancer Type**	**Outcomes**	Study Results
**Survival**			
Brandão et al., 2020 [55]	Breast cancer	(1) 3-year survival (2) 3-year survival—disease-free stage 0–III (3) 3-year survival—overall stage 0–III (4) Survival duration—stage IV	(1) 44.8% vs. 62.6% *p* = 0.039 (2) 41.7% vs. 56.8% *p* = 0.103 (3) 48.0% vs. 73.0% *p* = 0.003 (4) 19.4 months vs. 13.6 months (median) *p* = 0.059
Bydder et al., 2009 [39]	NSCLC	(1) 1-year survival (2) Survival duration	(1) 18% vs. 33% (2) 208 days vs. 237 days (median), 205 days vs. 280 days (mean), ***p* = 0.048**
Chen et al., 2018 [18]	CRA-LLM	(1) 1-year survival (2) 3-year survival (3) 5-year survival (4) 5-year survival—curative treatment	(1) 53.45% vs. 74.52% ***p* < 0.001** (2) 24.21% vs. 48.75% ***p* < 0.001** (3) 17.41% vs. 44.32% ***p* < 0.001** (4) 41.31% vs. 64.57% *p* = 0.062
Foucan et al., 2020 [37]	Colon cancer	Survival duration	After diagnosis 264.9 days vs. 338.0 days ***p* = 0.014** After surgery 252.1 days vs. 312.2 days *p* = 0.068
Hung et al., 2020 [46]	Stage III NSCLC	Survival duration	25.7 months vs. 41.2 months, ***p* = 0.018** (median)
Lan et al., 2016 [23]	Colorectal cancer	(1) 3-year survival (2) 3-year survival—liver metastasis (3) 3-year survival—lung metastasis	(1) 25.4% vs. 38.2% ***p* < 0.001** (2) 22.3% vs. 32.9% ***p* < 0.001** (3) 24.5% vs. 42.6% ***p* < 0.001**
MacDermid et al., 2009 [24]	Colorectal cancer	(1) 3-year survival—Dukes B (2) 3-year survival—Dukes C (3) Survival duration—surgical treatment for metastatic disease	(1) 76% vs. 70% *p* = 0.486 (2) 58% vs. 66% ***p* = 0.023** (3) 8 months vs. 11.9 months; *p* = 0.234 (median)
Munro et al., 2015 [25]	Colorectal cancer	(1) 5-year survival (2) 5-year survival—cause-specific (CSS) (3) 5-year survival—early disease (4) 5-year survival—advanced disease (5) survival—>6 weeks after diagnosis	(1) 33.6% vs. 52.3% ***p* < 0.001** (2) 48.2% vs. 63.1% ***p* < 0.001** (3) 86.4% vs. 80.6% *p* = 0.598 (4) 8.4% vs. 18.0% ***p* < 0.001** (5) 57.7% vs. 63.2% *p* = 0.064
Palmer et al., 2011 [27]	Rectal cancer	(1) 5-year survival (2) 5-year survival—resected without metastasis	(1) 28% vs. 30% (2) 52% vs. 34%
Pan et al., 2015 [41]	Non-small cell lung cancer	2-year survival—stage-specific	Stage I 78% vs. 81% Stage II 59% vs. 64% Stage III 31% vs. 37% Stage IV 20% vs. 22%
Stone et al., 2018 [43]	Lung cancer	Survival probability—HR	0.54 95% CI 0.45–0.65 ^a^ ***p* < 0.001** 0.70 95% CI 0.58–0.85 ^b^ ***p* < 0.001**
Tamburini et al., 2018 [44]	Non-small cell lung cancer	Survival—OR	0.48 95% CI 0.25–0.92
Wille-Jørgensen et al., 2013 [34]	Rectal cancer	Survival	No significant differences in survival, *p* = 0.33
Yang et al., 2020 [57]	Breast cancer	(1) Survival (2) Survival—HR (3) Survival—disease-free (4) Survival—disease-free (HR) (5) Survival—disease-free per treatment (HR)	(1) 97.19% vs. 98.98% ***p* < 0.001** (2) 2.760, 95% CI 1.642–4.641 ^a^ ***p* < 0.001** 2.478, 95% CI 1.431–4.291b ***p* = 0.001** (3) 89.69% vs. 93.89% ***p* < 0.001** (4) 1.888 95% CI 1.451–2.456 ^a^ ***p* < 0.001** 1.813 95% CI 1.367-2.405 ^b^ ***p* < 0.001** (5) Chemotherapy 1.502, 95% CI 1.033–2.183 ^b^ *p* = 0.131 Radiotherapy 2.313, 95% CI 1.540–3.475 ^b^ *p* < 0.001 Endocrine therapy 2.482, 95% CI 1.560–3.947 ^b^ ***p* < 0.001** Targeted therapy 1.763, 95% CI 1.001–3.105 ^b^ *p* = 0.095
Ye et al., 2012 [35]	Colorectal cancer	(1) 1-year survival (2) 3-year survival (3) 5-year survival	(1) 94.5% vs. 95.8% (2) 75.7% vs. 87.1% (3) 62.4% vs. 79.1% ***p* = 0.015**
**Recurrence or Metastasis**			
Brandão et al., 2020 [55]	Breast cancer	Recurrence	29% vs. 18%, *p* = 0.07
Richardson et al., 2016 [28]	Rectal cancer	Recurrence	Local only 10% vs. 0% (MDTM year 1) vs. 0% (MDTM year 2) Distant only 5% vs. 0% (MDTM year 1) vs. 2% (MDTM year 2) Local and distant 5% vs. 0% (MDTM year 1) vs. 0% (MDTM year 2)
Tsai et al., 2020 [56]	Breast cancer	Recurrence—HR	0.84, 95% CI 0.70-0.99 ^b^ ***p* = 0.047**
Wille-Jørgensen et al., 2013 [34]	Rectal cancer	(1) Local recurrence (2) Distant metastasis	(1) 4% vs. 3% (NS) (2) 15% vs. 21% (NS)
Ye et al., 2012 [35]	Colorectal cancer	(1) Recurrence rate	(1) Lower tumor recurrence in the MDTM group: ***p* < 0.001**
**Mortality**			
Brandão et al., 2020 [55]	Breast cancer	(1) HR of death (2) HR for recurrence or death (3) Death	(1) Overall population 0.77 (95% CI, 0.49–1.19) ^b^ Stage 0–III 0.47 (95% CI, 0.27–0.81) ^b^ (2) Overall population 0.72 (95% CI, 0.46–1.13) ^b^ (3) 23% vs. 10%, *p* = 0.07
Chen et al., 2018 [18]	CRA-LLM	HR of death	1.949 95% Cl 1.299–2.923 ^a^ ***p* < 0.001** 0.403 95% CI 0.251–0.647 ^b^ ***p* < 0.001**
Lan et al., 2016 [23]	Colorectal cancer	Surgical mortality	4.9% vs. 2.5%, ***p* = 0.049**
Munro et al., 2015 [25]	Colorectal cancer	HR for death	0.53, 95% CI 0.40–0.69 ^a^ ***p* < 0.001** 0.73, 95% CI 0.53–1.00 ^b^ ***p* = 0.047**
Pan et al., 2015 [41]	Non-small cell lung cancer	(1) Death rate and HR of death—stage-specific	(1) Stage I–II 84.65% vs. 15.36% 0.89, 95% CI 0.78–1.01 ^b^ *p* = 0.060 Stage III–IV 85.97% vs. 14.03% 0.87, 95% CI 0.84–0.90 ^b^ ***p* < 0.001**
Tamburini et al., 2018 [44]	Non-small cell lung cancer	(1) Mortality	18% vs. 8%; *p* = 0.006
Tsai et al., 2020 [56]	Breast cancer	(1) Mortality (2) Risk of mortality—HR	(1) 13.05% vs. 12.48% (2) 0.89, 95% CI 0.82–0.96 ***p* = 0.004**
Wille-Jørgensen et al., 2013 [34]	Rectal cancer	Post-operative mortality	9% vs. 5% ***p* = 0.007**
**Other**			
Brandão et al., 2020 [55]	Breast cancer	(1) Clean surgical margins (2) Axillary surgery completeness: (a) Complete (b) incomplete	(1) 88.4% vs. 92.3% *p* = 0.575 (2) (a) 58.4% vs. 67.9% (b) 29.9% vs. 21.4% *p* = 0.423
Richardson et al., 2016 [28]	Colorectal cancer (rectal cancer)	(1) Time to recurrence (months) (2) Quality of surgery	(1) 27 months vs. 14.5 months (MDTM year 1) vs. 6.5 months (MDTM year 2) (2) Completeness of TME (Complete/Nearly complete) 6% vs. 61% (MDTM year 1) vs. 76% (MDTM year 2)
Swellengrebel et al., 2011 [31]	Rectal cancer	Positive circumferential resection margins rate	10% vs. 14% *p* = 0.392
Tamburini et al., 2018 [44]	NSCLC	(1) completeness of resection (2) postoperative complications	(1) 92.4% vs. 94.1% *p* = 0.52 (2) 40.6% vs. 40.0% *p* = 0.91
Ye et al., 2012 [35]	Colorectal cancer	Time to recurrence (months)	11.0 vs. 14.1 months ***p* < 0.001**

^a^ = univariate analysis, ^b^ = multivariate analysis. Results, if available, comparing patients in MDTM group and non-MDTM group, are shown as (‘results non-MDTM group’ vs. ‘results MDTM group’, *p*-value). CRA-LLM = colorectal cancer with lung or liver metastasis, HR = hazard ratio, MDTM = multidisciplinary team meeting, NSCLC = non-small cell lung cancer, OR = odds ratio. P-values < 0.05 were considered significant, annotated in bold.

**Table 6 cancers-13-04159-t006:** Summary of main outcomes per cancer type.

	Colorectal Cancer	Lung Cancer	Breast Cancer	Prostate Cancer
Process outcomes	Time to treatment: increased in MDTM group [26,37]	Time to treatment: increased in MDTM group [38,47], no effect [38], decreased in MDTM group [40] Costs: mean cost of care reduced in MDTM group [40] Other: increased guideline adherence and research participation in MDTM group [40]	Time to treatment: no effect [55] Costs: MDTM is cost-effective [55]	N/A
Proportion of cases with changed overall management plans in %	Range: 6–29% Weighted average: 16.2% [19,20,21,22,30]	Range: 53–58% Weighted average: 53.2% [42,45]	Range: 42.1% Weighted average: - [54]	Range: 1.6–43% Weighted average: 27.1% [48,49,50,51,52,53]
Diagnostics, treatments and palliative care	MRI: increase in MDTM group [17,20,36] CT: increase in MDTM group [17,20,35,37], no effect [20,36] US: no effect [17,20,36] Colonoscopy: no effect [20,28,36] Surgery: reduced in MDTM group [23,37] Surgery type: significant effect [27,28,33] Radiotherapy: increase in MDTM group [23], no effect [24] Chemotherapy: increase in MDTM group [23], decrease or no effect in MDTM group [35] Palliative care: increase in MDTM group [23], no effect [24]	Diagnostics: N/A Surgery: no effect [38], reduced in MDTM group [40] Surgery type: significant effect [44] Radiotherapy: increase in MDTM group [38], no effect [39] Chemotherapy: increase in MDTM group [38], no effect [39] Palliative care: increase in MDTM group [38,40], no effect [39,43]	Diagnostics: N/A Surgery: no effect [55] Surgery type: no effect [55] Radiotherapy: no effect [55] Chemotherapy: no effect [55] Palliative care: no effect [56]	N/A
Patient outcomes	Survival: improved in MDTM group [18,23,24,25,35,37], no effect [34]	Survival: improved in MDTM group [39,41,43,44,46]	Survival: improved in MDTM group [55,56,57]	N/A

Overview of the main outcomes per cancer type as described in the results section. Increase/decrease in MDTM group = a significant increase or decrease of the occurrence of that specific outcome was measured in the MDTM group compared to the control. Significant effect = significant differences were identified between the MDTM group and control. No effect = no significant differences were identified between MDTM group and control. MDTM = multidisciplinary team meeting.

## Data Availability

Data sharing is not applicable to this article as no new data were created or analyzed in this study.

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
