# Peer review of "The Effects of Multidisciplinary Team Meetings on Clinical Practice for Colorectal, Lung, Prostate and Breast Cancer: A Systematic Review"

_cancers, 2021, doi:10.3390/cancers13164159_

Round 1

Reviewer 1 Report

This is a metanalysis on  the effects of multidisciplinary  team meetings (MDTM) for lung, breast, colorectal and prostate cancer.

The paper is well written , according to the metanalysis guidelines.

I think the paper can be considered for publication.

Author Response

Response to Reviewer 1 comments:

Point 1: This is a metanalysis on the effects of multidisciplinary team meetings (MDTM) for lung, breast, colorectal and prostate cancer.

The paper is well written, according to the metanalysis guidelines.

I think the paper can be considered for publication.

Response 1:

Thank you for the evaluation of our manuscript.

Reviewer 2 Report

This manuscript was well written based on a previous outcomes accroding to specicic cancer site,

I woudl like to address a couple of questions,

  1. Why did you want to target these organ cancer about systemic review about MDT
  2. the clinical impact of MDT on clincal outcomes and cost for 5 cancer , it seems to be difficult to follow.
  3. Table s are too many and huge and busy
  4. Please to focus the specific cancer organ and raise the issues about current clincal pracitce,.
  5. The cost effectiveness and other intervening factors of MDT should be addressed in detail.

Author Response

Response to Reviewer 2 comments:

Point 1: Why did you want to target these organ cancer about systemic review about MDT

Response 1:

We targeted these four cancer types for the following reasons. Colorectal, lung, prostate and breast cancer are the top four cancer types when we look at worldwide incidence. It is also known that cancer incidence for these cancer types are increasing every year. This can potentially lead to an increased workload of clinicians, which also relates to MDTMs. Since these are already considered time-consuming it is important to understand how they impact the different aspects of the clinical pathway and patient outcomes.

For clarity we have referred to the four cancer types in the sentence where we state we want to focus on the cancer types which are expected to have high impact on global healthcare:

‘The total workload of clinicians, occupied by MDTMs, is expected to increase especially for cancer types with continuously increasing incidence rates such as colorectal, lung, prostate and breast cancer [4].

These cancer types together constitute the top four cancer types in terms of global annual incidence [5]. Therefore, it is of great importance that the impact of MDTMs on different aspects of the clinical pathway and patient outcomes are well understood. […] We are the first to report on multiple cancer types in detail, to compare the effects of MDTMs in the four cancer types (colorectal, lung prostate and breast cancer) that are expected to have a high impact on global healthcare.’

Point 2: the clinical impact of MDT on clincal outcomes and cost for 5 cancer , it seems to be difficult to follow.

Response 2:

As described above, our manuscript discusses the value of MDTM’s in the four most common and impactful types of cancer. We discussed different types of impact for different cancer types in the various sections. To increase clarity of the manuscript we also added a requested summary table.

Point 3: Table s are too many and huge and busy

Response 3:

There are indeed 5 tables included in our manuscript, of which the format has been chosen with the following careful considerations:

  • There are many studies, measuring very similar, but not always exactly the same outcomes.
  • In order to facilitate as much comparability as possible between studies per outcome measure, we have decided to divide the results of all included studies over 4 different tables (Tables 2-5). Putting all results per study in one table would be unclear and difficult to interpret.
  • We have condensed the results from each study to the main outcome measures as much as possible. This way the differences and similarities between studies can be identified more easily.
  • We use the same format to show results within each table to ensure the information can be interpreted without too much difficulty.
  • We have chosen this approach to report the results in the tables, so readers are able to gain an overview of the available knowledge from the included articles without having to access them all individually.
  • In addition, we added a summary table, as requested by reviewer 3.

Point 4: Please to focus the specific cancer organ and raise the issues about current clincal pracitce,.

Response 4:

In the discussion section we have mentioned differences between the included cancer types where possible (changes in management plans). Heterogeneity of data and lack of a meta-analysis makes it difficult to say anything about differences between cancer types with full certainty. For that reason, we have chosen the current approach.

Point 5: The cost effectiveness and other intervening factors of MDT should be addressed in detail.

Response 5:

Cost-effectiveness and intervening factors are very interesting outcomes to address indeed. However, only three articles within the scope of our review have studied cost-related outcomes, using different approaches and different settings and are thus not directly comparable. We cannot really conclude anything about cost-effectiveness based on this limited information. For that reason, we have suggested more research on costs and cost-effectiveness of MDTMs is necessary in the ‘future research’ section of our discussion.

Reviewer 3 Report

Cancers 

The effects of multidisciplinary team meetings on clinical practice for colorectal, lung, prostate and breast cancer: a systematic review

Dr. Koco and Weekenstroo's team worked on a systematic review of the impact of MDT meetings on the management of the four most prevalent cancers across the world.  The authors should be congratulated for this difficult work because of the heterogeneity of practices and the low level of evidence, which did not include any randomized trials. They have carefully analyzed a large amount of literature in an attempt to show a trend of effect on the choice of type of surgery, a better preoperative radiological workup, and a trend towards greater survival, at the cost of time and money, given an increasing incidence of the cancers investigated.

Here are some remarks on the manuscript that could improve its scope:

Major:

  • Introduction l.65-66: your introduction should be meant as the time of setting the background of your study and the aim of your work. At this point, talking about your results and arguing the prominence of your positive results among the current literature should be reserved for your discussion and conclusion.
  • Methods: the exclusion of 13,081 records is not well explained. The authors may address this with a more specific explanation of “title-abstract screening”. The flowchart may be more transparent with the exclusion reasons.
  • Results: a synthetic summary table might be necessary for more clarity about all the outcomes of interest which are discussed.
  • Discussion: “Strengths and limitations”; arguing about the study’s strengths should stay objective and consistent with the actual description of methodology and description. “13,000” seems a little bit misleading about the real number of full-text articles (165) that were assessed for eligibility and eventually reviewed by the authors. This point should be consequently addressed for more transparency.

Again, I enjoyed reading this important manuscript!

Author Response

Please see the attachment for point-by-point response to the reviewer's comments.
